# Humoral and Cellular Immunity following Five Doses of COVID-19 Vaccines in Solid Organ Transplant Recipients: A Systematic Review and Meta-Analysis

**DOI:** 10.3390/vaccines11071166

**Published:** 2023-06-27

**Authors:** Abdulmalik S. Alotaibi, Heba A. Shalabi, Abdullah A. Alhifany, Nouf E. Alotaibi, Mohammed A. Alnuhait, Abdulrahman R. Altheaby, Abdulfattah Y. Alhazmi

**Affiliations:** 1Clinical Pharmacy Department, College of Pharmacy, Umm Al-Qura University, Makkah 21955, Saudi Arabia; dr.heba.pharmd@gmail.com (H.A.S.); aahifany@uqu.edu.sa (A.A.A.); nealotaibi@uqu.edu.sa (N.E.A.); manuhait@uqu.edu.sa (M.A.A.); ayhazmi@uqu.edu.sa (A.Y.A.); 2Organ Transplant Center, King Saud bin Abdulaziz University for Health Sciences, Riyadh 11426, Saudi Arabia; a83sa@hotmail.com

**Keywords:** solid organ transplant, immunosuppressed patients, humoral and cellular immunity, COVID-19 vaccine, BNT162b2 (BioNTech/Pfizer), kidney transplant, liver transplant, heart transplant, lung transplant

## Abstract

Solid organ transplant (SOT) recipients are at increased risk of COVID-19 infection because of their suppressed immunity. The available data show that COVID-19 vaccines are less effective in SOT recipients. We aimed to assess the cellular and humoral immunogenicity with an increasing the number of doses of COVID-19 vaccines in SOT recipients and to identify factors affecting vaccine response in this population. A systematic review and meta-analysis were conducted to identify ongoing and completed studies of humoral and cellular immunity following COVID-19 vaccines in SOT recipients. The search retrieved 278 results with 45 duplicates, and 43 records did not match the inclusion criteria. After title and abstract screening, we retained 189 records, and 135 records were excluded. The reasons for exclusion involved studies with immunocompromised patients (non-transplant recipients), dialysis patients, and individuals who had already recovered from SARS-CoV-2 infection. After full-text reading, 55 observational studies and randomized clinical trials (RCTs) were included. The proportion of responders appeared higher after the third, fourth, and fifth doses. The risk factors for non-response included older age and the use of mycophenolate mofetil, corticosteroids, and other immunosuppressants. This systematic review and meta-analysis demonstrates the immunogenicity following different doses of COVID-19 vaccines among SOT patients. Due to the low immunogenicity of vaccines, additional strategies to improve vaccine response may be necessary.

## 1. Introduction

COVID-19 or the coronavirus-associated disease of 2019, caused by the novel coronavirus SARS-CoV-2, affected the field of solid organ transplantation profoundly, which caused a decline in transplant activity during the pandemic’s first wave [1]. In addition, the number of comorbidities these individuals have has been linked to a poorer response to the COVID-19 vaccine rather than the net state of immunosuppression [2,3]. Solid organ transplant (SOT) recipients were a priority for immunization against COVID-19 because they were developing a higher rate of COVID-19-related complications. Additionally, all of the COVID-19 vaccinations were suitable and safe for transplant recipients because none of them used live, replicating viral vectors [4].

There is a paucity of clinical effectiveness studies of SARS-CoV-2 vaccines in the setting of SOT. Antibody (humoral immunity) responses to COVID-19 vaccines in SOT recipients are diminished compared with those of the general population [5]. A retrospective study reported that the anti-SARS-CoV-2 antibody prevalence was 0% prior to the first dose, 4% prior to the second dose, 40% prior to the third dose, and 68% four weeks after the third dose in SOT recipients; nevertheless, the number of doses of COVID-19 vaccines recommended for SOT recipients is still not clear [6]. On the other hand, a study investigated T cell (cellular immunity) responses in SOT recipients and reported that the percentage of CD4+ T cells responding to the second dose of the vaccine was reported to be 90% [7]. The relationship between these T cell responses and protection against SARS-CoV-2 infection or a decrease in illness severity is still unknown, in contrast to that for antibody responses [8].

Hence, our study aimed to assess the effect of different doses of COVID-19 vaccines on the immune responses, both cellular and humoral, in SOT recipients and identify the factors affecting vaccine response in this population.

## 2. Materials and Methods

The Preferred Reporting Items for Systematic Reviews and Meta-Analyses (PRISMA) criteria were followed for this systematic review and meta-analysis [9]. This systematic review was registered in the PROSPERO database, and the registration number is CRD42022340783.

### 2.1. Search Strategy

A comprehensive systematic literature search was performed utilizing the databases PubMed, Scopus, ClinicalTrials.gov, and Web of Science to identify the ongoing and completed clinical trials of humoral and cellular immunity following COVID-19 vaccines in SOT recipients, and it retrieved all studies in the English language identified from inception to 1 January 2023 (see Appendix A).

We implemented a two-stage screening process, initially by evaluating titles and abstracts, followed by a thorough examination of full-text articles. Two researchers, H.S. and A.S., independently screened each title, abstract, and full text. Any discrepancies that arose were resolved through reaching a consensus with a third researcher, A.Y.

We performed a meta-analysis of observational studies and randomized clinical trials (RCTs) that met the following inclusion criterion: SOT recipients over 18 years old who received one, two, three, four, or five doses of any type of COVID-19 vaccine. RCTs, cohorts, case series, and case reports were included.

We excluded studies that enrolled immunocompromised patients who were not transplant recipients, dialysis patients, individuals who had previously recovered from SARS-CoV-2 or were actively infected at the time of vaccination or within seven days after the second dose. Additionally, patients who tested positive for SARS-CoV-2 in polymerase chain reaction (PCR) tests before or after receiving the first dose and/or within the first week following the second dose were also excluded.

### 2.2. Data Extraction

Two authors (H.S. and A.S.) utilized a standardized data extraction form to extract the data. The information included the first author, publication date, study design, type of vaccination, number of transplant recipients, type of transplant, number of healthy controls (HCs), seroconversion, and factors influencing vaccine response. The extracted key data underwent a thorough review, and their quality was assessed in the conclusion of the data extraction process by the same two researchers (refer to Appendix A).

### 2.3. Risk of Bias Assessment

The Newcastle–Ottawa Scale (NOS) evaluates the risk of bias in non-randomized trials. The selection of the research group, group comparability, and the confirmation of either the exposure or the result of interest for case–control or cohort studies, respectively, were utilized as the three key aspects in the “star system” used to rate studies. Each numerical item in the selection and exposure categories can earn a study a maximum of one star. For comparability, a maximum of two stars may be assigned.

Two reviewers (H.S. and A.S.) independently judged these domains, and all discrepancies were resolved by obtaining the independent opinion of a third reviewer (A.Y.).

Randomized clinical trials were assessed using Review Manager 5.4 software. The desktop version of the program was used for non-Cochrane reviews, offline work, and altering reviews that were not currently editable in RevMan Web (diagnostic test accuracy reviews) (see Appendix A).

## 3. Results

The search retrieved 278 results on PubMed/Medline, with 45 duplicates being excluded. After title and abstract screening, of the 233 records, 178 records were excluded. After reading the full texts, 55 studies were included [6,10,11,12,13,14,15,16,17,18,19,20,21,22,23,24,25,26,27,28,29,30,31,32,33,34,35,36,37,38,39,40,41,42,43,44,45,46,47,48,49,50,51,52,53,54,55,56,57,58,59,60,61] (Figure 1). Many studies were prone to bias; two of them were RCTs estimated to be of good quality and the others were observational studies (*n* = 55).

The review included 55 studies, of which 52 were analyzed in the meta-analysis aiming to assess the humoral response in accordance with each additional vaccine dose, with 7303 patients being included overall; 44.5% (3251 patients) achieved a humoral response following the administration of all doses of vaccines, and the random effect model was 0.44 (95% confidence interval = 0.39 to 0.49; I^2^ = 93.7%). (See Appendix A).

The humoral immune response was identified using the cutoff value provided by the assay’s manufacturer to detect the presence of anti-spike antibodies. These antibodies specifically target the spike protein of SARS-CoV-2, and include anti-RBD or anti-S1 antibodies. On the other hand, the cellular immune response in the studies was determined by assessing the presence of SARS-CoV-2-specific T cells. This assessment was conducted using techniques such as T-EliSpot, interferon-γ release assays, or the detection of activation-induced markers (AIM) in flow cytometry-sorted cells. These methods allowed the researchers to identify and measure the activation or response of T cells specifically targeting SARS-CoV-2.

### 3.1. Cellular and Humoral Responses after a First Dose

As shown in Appendix A [14,19,21,60], studies conducted on a variety of SOT recipients explain the immunogenicity of the COVID-19 vaccines, which were developed by Pfizer-BioNTech (BNT162b2), Moderna (mRNA-1273), Oxford/AstraZeneca (ChA-dOx1 nCoV-19), and Janssen (Ad26.COV2. S).

The cellular response following the first dose of the COVID-19 vaccine among SOT recipients was extremely low compared to that of the general population. Hall et al. observed that only 4 out of 40 participants (10%) developed a T cell response following the first dose of the vaccine. Gianpiero et al. showed poor T cell responses following the first dose of the vaccine in liver transplant (LT) recipients, where the median IFN-γ level increased from 3.5 pg/mL (interquartile range (IQR) of 0.1–10.8) to 9.5 pg/mL (IQR of 2.1–29.9), whereas in HCs, the median IFN-γ level was 0.85 pg/mL (IQR of 0.1–5.7), but it increased after the first dose to 112.2 pg/mL (IQR of 53.5–205.0).

The humoral response following the first dose also showed a poor antibody response among SOT recipients. One study by Benotmane et al. showed that the seroconversion rate was 10.8% after the first dose of the Moderna mRNA-1273 COVID-19 vaccine [19]. Only 2 out of 12 individuals who received the Janssen vaccination were shown to have anti-RBD antibodies, according to Boyarsky, as opposed to the 430 out of 725 patients who finished the mRNA vaccine series (17% versus 59%) [15]. According to Schmidt et al.’s findings, 95.7% of immunocompetent individuals and 26.3% of transplant recipients exhibited detectable antibodies and/or T cells following a single dose of mRNA or vector-based vaccine [14].

In this meta-analysis, 17 out of the 52 included studies reported a humoral response after the first dose, which showed that 850 patients (22.7%) out of 3735 patients developed humoral responses, with a pooled proportional random effect of 0.12 (95% CI = 0.07 to 0.2, I^2^ = 81.6%) (see Figure 2).

### 3.2. Cellular and Humoral Responses after a Second Dose 

Our review of the literature revealed 37 studies that showed COVID-19 vaccine effectiveness after only two doses. Most of these studies pertained to vaccines that were developed by Pfizer-BioNTech (BNT162b2), Moderna (mRNA-1273), or Oxford/AstraZeneca (ChAdOx1 nCoV19) [12,13,15,16,17,18,19,20,22,23,29,30,31,32,34,37,40,42,43,44,45,46,47,48,49,50,51,52,53,54,55,56,57,58,59,60,61].

In June 2021, Brian J. Boyarsky et al. demonstrated slight improvements in the cellular responses of transplant recipients (n = 658) after a second dose compared to those after a first. Nearly 51% of the study population received the BNT162b2 (Pfizer-BioNTech) vaccine and 49% received the mRNA1273 (Moderna) vaccine. A detectable antibody response was observed in 98 participants (15%) following dose one and in 357 participants (54%) after dose two at a median (IQR) of 29 (28–31) days. Overall, a majority of the participants exhibited detectable antibody responses after receiving the second dose. However, those who did not show a response after the first dose had relatively low levels of antibodies [58].

Victoria G. Hall et al. reported on the humoral response in (n = 127) SOT recipients. Anti-RBD antibodies were detected in 34.5% of the participants after the second dose of the mRNA-1273 (Moderna) vaccine, and 26.9% developed neutralizing antibodies primarily after the second dose [55]. In July 2021, in a single-center cohort study conducted by Alessandra Mazzola et al., the humoral response to the BNT162b2 vaccine (Pfizer/BioNTech) was evaluated in SOT recipients (n = 143) and HCs (n = 25), with the seroconversion rate after the second dose being significantly lower in the SOT recipients compared to that in the HCs (28.6% vs. 100.0%, respectively; *p* < 0.0001) [40].

The meta-analysis reported on the seroconversion rates in 29 studies with a total of 3433 patients, of whom 1180 (34.3%) exhibited a humoral response after receiving a second dose of the COVID-19 vaccine, with a pooled proportion of the random effects model of 0.3 (95% CI = 0.26 to 0.34; I^2^ = 81.6%). (see Figure 3).

### 3.3. Cellular and Humoral Responses after a Third Dose 

The cellular response following the third vaccination was demonstrated in a RCT where only 17 patients out of 197 developed a T cell response, whereas an antibody response was found in 39% of the study population [25]. In a separate RCT, 120 SOT recipients who received two doses of mRNA-1273 were randomly assigned to receive a third dose or a placebo two months later (dosing schedule: 0, 1, and 3 months). Hall et al. found that 55% of the mRNA-1273 group and 18% of the placebo group achieved a serological response characterized by an anti-RBD antibody level of at least 100 U per milliliter. The relative risk was 3.1 (95% CI, 1.7 to 5.8; *p* < 0.001). The median virus neutralization rate after the third dose was 71% in the mRNA-1273 group and 13% in the placebo group. Additionally, after receiving the third dosage, the mRNA-1273 group had higher median SARS-CoV-2-specific T cell counts (432 vs. 67 cells per 106 CD4+ T cells; 95% CI for the between-group difference: 46 to 986) than the placebo group did [33].

In a case series of 30 SOT recipients, William A. Werbel et al. reported that after receiving a third dose of the Janssen, Moderna (mRNA1273), or Pfizer-BioNTech vaccine, there was a notable increase in positive antibody titers. Six patients who initially had low-positive antibody titers exhibited high-positive antibody titers following the third dose. Of the 24 patients who initially had negative antibody titers before the administration of the third dose, only 25% exhibited high-positive antibody titers after receiving the third dose. Additionally, 8% of the patients showed low-positive antibody titers, while the remaining 67% maintained negative antibody titers [9].

Nassim Kamar et al. studied the prevalence of anti-SARS-CoV-2 antibodies among 101 different SOT recipients, and the results showed that prior to the first dose, 0% (95% CI, 0 to 4; 0/101) of the SOT recipients had detectable antibodies, but the proportion increased to 4% (95% CI, 1 to 10; 4/101 patients) prior to the second dose and 40% (95% CI, 31 to 51; 40/99) prior to the third dose. Four weeks after the third dose, 68% (95% CI, 58 to 77; 67/99) of the SOT recipients had detectable antibodies. A third dose of the BNT162b2 vaccine resulted in improved immunogenicity. Notably, no cases of COVID-19 were reported among these patients.

Antibody responses from administering a third dose of vaccination were reported in 12 studies with a total of 816 patients, of whom 350 (42.9%) achieved a humoral response. The random effect model was 0.43 (95% CI: 0.34 to 0.52; I^2^ = 83.3%). (see Figure 4).

### 3.4. Cellular and Humoral Responses after a Fourth Dose 

In January 2022, Sophie Caillard et al. investigated whether or not administering an mRNA-based anti-SARS-CoV-2 vaccine to KT recipients who had minimal serologic responses to the previous three doses would result in an increase in anti-spike IgG titers. Therefore, one month after a third dose, 92 KT recipients with anti-spike IgG titers of less than 143 BAU/mL were studied; 34 individuals received a fourth dose of the BNT162b2 mRNA vaccine (Pfizer), while 58 individuals received a fourth dose of the mRNA-1273 vaccine (Moderna). The data showed that the proportion of patients who had anti-spike IgG titers above 143 BAU/mL following the fourth dose of the BNT162b2 vaccine was 48%, and there were higher IgG titer levels for those who received the mRNA-1273 vaccine (52%) [38].

Nassim Kamar et al. assessed the efficacy of a fourth dose of the mRNA-based BNT162b2 vaccine (Pfizer-BioNTech) in 37 solid organ transplant (SOT) recipients. Among them, 13.5% had a weak response to the previous three doses, and 83.8% had no response. Prior to the fourth dose, only 13.5% of the patients had detectable anti-SARS-CoV-2 antibodies. However, one month later, 18 out of 37 patients (48.6%) had detectable antibodies (*p* = 0.002). Among the five patients who had seropositive results before the fourth dose, the median antibody concentration increased significantly from 4 BAU/mL (range of 1–9 BAU/mL) to 402 BAU/mL (range of 87–508 BAU/mL) four weeks after the fourth dose (*p*  <  0.001). When compared to individuals who had no response, those with detectable antibodies before the fourth dose had significantly higher antibody concentrations 4 weeks after dose four was administered [28].

Out of a total of 172 patients, 84 (48.8%) showed antibody responses after receiving four doses of the vaccine, with an I^2^ of 84.8% and a pooled proportion random effect model of 0.48 (95% CI: 0.25 to 0.72). (see Figure 5).

### 3.5. Cellular and Humoral Responses after a Fifth Dose

Aura T. Abedon et al. described a case series in which SOT recipients showed enhanced antibody responses after receiving dose five of a SARS-CoV-2 vaccination. Of the 18 participants who received a fifth dose, 56% had previously receive1d four doses of an mRNA vaccine, while 44% had received three doses of a mRNA vaccine and one dose of Ad.26.COV2.S. Antibody testing was conducted using the Roche enzyme immunoassay between 21 and 34 days after the fifth dose. Of the 17 participants who were tested on the same platform, all had higher antibody titers (100%). However, one participant tested seronegative on two different platforms before and after receiving the fifth dose. The study’s authors highlighted the importance of continuing to test for antibodies even after receipt of a booster dose and considering other approaches such as passive immunoprophylaxis or immunosuppressive modulation for those who do not respond well to vaccination. In May 2022, a cohort study analyzed a total of 4277 SARS-CoV-2 vaccinations in 1478 patients. Following 1203 basic vaccinations, the serological response rate was 19.5%. However, for subsets of 603, 250, and 40 patients, the response rates increased to 29.4%, 55.6%, and 57.5% after the third, fourth, and fifth vaccinations, respectively. The cumulative response rate was found to be 88.7% [61].

### 3.6. Immunogenicity in the Organ-Transplanted Population

LT recipients receive less intense induction and maintenance immunosuppressive medications, potentially leading to a more robust humoral response. In a study by Alexandra T. Strauss et al., it was determined whether or not LT recipients who received a two-dose mRNA vaccine series, either mRNA-1273 (Moderna) or BNT162b2 (Pfizer-BioNTech), developed SARS-CoV-2 antibodies. With a median of 21 days (IQR of 19–25 days) after the first dose, the study found that 34% (95% CI, 27–42%) of the participants exhibited detectable antibodies and 81% (95% CI, 74–87%) after the second dose at a median of 30 days (IQR of 28–31 days) as shown in Figure 2, and 3. Among these individuals, 34% were responders to the priming dose (D1+/D2+), 47% were responders to the booster dose (D1−/D2+), and 19% did not respond to the vaccine [22]. Rabinowich L. et al. reported that 80 LT patients in a cohort that had received two doses of the BNT162b2 vaccination had a lower antibody response rate of only 47.5%, which was in contrast to the findings of Alexandra T. Strauss et al.’s study [16]. In October 2021, Gianpiero D’Offizi et al. reported that the humoral response, measured as the anti-spike antibody level, was significantly lower in LT recipients compared to that of the HCs after receiving the second dose of the anti-SARS-CoV2 mRNA vaccine (77.0% vs. 100%, *p* = 0.001). On the other hand, the cellular response data showed that the LT recipients had a significantly lower positive IFN-γ response at dose two compared to the HCs (72.1% vs. 100%, *p* < 0.0001). In both the humoral and cellular arms, the response rates for the LT recipients were lower than those for the HCs [49]. Yana Davidov and colleagues conducted another study that found that LT recipients had lower immune responses to the BNT162b2 mRNA vaccine than the HC group did, and they were of similar ages. The study showed that out of 76 liver transplant recipients, 55 (72.4%) had a positive antibody response, while out of 174 immunocompetent controls, 164 (94.3%) had a positive response (OR, 6.26; 95% CI, 2.8–14.1; *p* < 0.0001), as measured at a median 35 days (IQR of 17–52 days) after the second vaccine dose [51].

Moreover, kidney transplant (KT) recipients showed inadequate antibody responses to SARS-CoV-2 mRNA vaccinations. In a prospective cohort study of consecutive KT recipients, after receiving the second dose of the BNT162b2 vaccine, only 36.4% of the 308 KT recipients tested positive for anti-spike (anti-S) antibodies within 2–4 weeks [12]. Grupper et al. reported poor humoral responses to two doses of the BNT162b2 SARS-CoV-2 vaccine in KT recipients. The study included 136 KT recipients who had received the full vaccination, and it compared their humoral responses to those of 25 HCs. The outcomes demonstrated that all HCs generated positive responses to the spike protein, whereas only 51 of the 136 transplant recipients (37.5%) had positive serology, demonstrating a statistically significant difference (*p* = 0.001) [15]. In contrast, Korth J. et al. reported that only 5 of 23 (22%) KT recipients tested positive for SARS-CoV-2 IgG antibodies vs. 23 (100%) HCs after a second dose of vaccine of BNT162b2 (Pfizer-BioNTech) [17]. While Rezzan Eren Sadio˘glu et al. noticed that the antibody responses after two doses of inactivated vaccine (CoronaVac) among 85 KT recipients were considerably low (18.8%) [31], Louise Benning et al. reported that among 173 KT recipients who received two doses of mRNA or AstraZeneca vaccines, anti-spike1, anti-receptor binding domain, and surrogate neutralizing antibodies were detectable in 30%, 27%, and 24%, respectively [32].

Moreover, the immune responses of 77 heart transplant (HT) recipients who received two doses of the BNT162b2 vaccine were reported in a prospective study by Yael Peled et al. Anti-RBD IgG antibodies were detectable in 14 (18%) of the recipients after a mean of 21 days following the second dose. Of those with IgG anti-RBD antibodies, eight (57%) had immune sera-neutralized SARS-CoV-2 pseudo-virus [44]. The same author conducted another cohort study which found that among 96 adult HT recipients, a homologous third dose of the Pfizer BNT162b2 vaccine administered 18 days after the second dose resulted in a significant increase in the positive antibody response from 23% to 67%, along with a corresponding increase in neutralizing capacity. The third dose induced SARS-CoV-2 neutralization titers greater than nine times and IgG anti-RBD antibodies greater than three times the range achieved after the two primary doses. [26]. In contrast, Itzhaki Ben Zadok et al. found that HT recipients had lower rates of positive S-IgG antibody titers after their first BNT162b2 (Pfizer-BioNTech) vaccine dose, with only 15% demonstrating the presence of these antibodies (GMT 90 (IQR of 54–229) AU/mL). The HT recipients had an overall S-IgG antibody induction rate of 49% in response to either one or two doses of vaccine (GMT 426 (IQR 106–884) AU/mL) [13].

### 3.7. Homologous and Heterologous COVID-19 Vaccines

Our literature search resulted in three studies that assessed immunogenicity after the administration of homologous and heterologous COVID-19 vaccines [41,42,56]. Ana Luísa Correia et al., in January 2022, argued that the serological response in KT recipients may be influenced by the type of SARS-CoV-2 vaccine administered. The study included 131 kidney transplant (KT) patients who had received either two doses of an mRNA vaccine (BNT162b2 from [Pfizer-BioNTech, United States (US)-Germany] or mRNA-1273 from [Moderna, US]) or two doses of replication-deficient adenovirus vector-based vaccines (ChAdOx1-S from [Oxford, AstraZeneca, United Kingdom] or Ad26.CoV2.S from [Johnson & Johnson, US]). According to the study results, there were significant differences in the serological responses between the patients who received mRNA vaccines and those who received adenovirus vector vaccines, with higher rates of seroconversion observed in the mRNA vaccine group (67% vs. 33%, respectively; *p* < 0.001). Additionally, the mRNA vaccine group had higher levels of anti-spike IgG titers [41].

On the other hand, in December 2021, Çiğdem Erol et al. concluded that there were no significant seropositive differences between transplant recipients who received second doses of an inactivated whole-virus SARS-CoV-2 vaccine (Sinovac, Beijing, China) versus those who received the BNT162b2 vaccine (BioNTech/Pfizer) [42].

In KT recipients who failed to develop antibodies against the SARS-CoV-2 spike protein after receiving two doses of a mRNA vaccine, Roman Reindl-Schwaighofer and colleagues conducted a RCT in December 2021 to compare the effectiveness of a third dose of an mRNA vaccine versus a third dose of a vector vaccine. Out of the total study population of 197 KT recipients, 39% of them developed SARS-CoV-2 antibodies after receiving the third dose of an mRNA vaccine or the third dose of a vector vaccine. With anti-body response rates for the mRNA and vector vaccinations, respectively, of 35% and 42%; the study found no statistically significant difference between the groups. The T cell responses assessed via IGRA were also weak, with only 17 patients exhibiting positive responses after receiving a third vaccination with either an mRNA vaccine or a vector vaccine. In addition, only 22% of the patients who had seroconverted had neutralizing antibodies [25].

### 3.8. Factors That Affect the Immune Response to Vaccination 

Various risk factors affect the humoral and cellular responses among the SOT population, including the estimated glomerular filtration rate (eGFR), older age, time from transplantation, and type of immunosuppression regimen. (See Table 1, and Appendix A).

#### 3.8.1. Age

In the studies included in the meta-analysis to measure the factors that may have affected the seroconversion rates and the development of humoral responses, age appeared to be a significant factor affecting humoral response, as shown in 29 studies (n = 3572), where 1501 patients’ results (42%) indicated that a younger age significantly improved the response to the vaccine (mean difference (MD) = −4.364; 95% CI = −5.67 to −3.06; *p* < 0.001). The mean age of the vaccine responders was 4.364 years younger than that of the non-responders. A lower age was associated with a higher vaccine response. These results showed moderate heterogeneity (*p* = 0.006, I^2^ = 44%).

#### 3.8.2. Antimetabolites

Among 36 studies, where n = 4971 patients, 1073 patients (21.5%) were responders, and the pooled mean difference showed that using less antimetabolites significantly improved the response to a vaccine (OR = 0.238, *p* < 0.001). Those who showed a response to a vaccine received 76.6% less antimetabolite than those who did not show a response.

#### 3.8.3. Antithymocyte Globulin

Six studies considered anti-thymocyte globulin exposure in a total of 1214 total patients. The weighted odds ratio of all six studies was 0.847, showing that anti-thymocyte globulin exposure had no significant effect on the pooled odds of the vaccine immune response (pooled *p* = 0.4).

#### 3.8.4. Body Mass Index (BMI)

The weighted mean difference of all 15 studies that took BMI into consideration was -0.469, with a pooled standard error of 0.219, and 994 (43.5%) responders out of a total of 2283 patients had a 0.469 lower-pooled BMI mean difference compared to the non-responders (pooled *p* = 0.032). A lower BMI was associated with a higher vaccine response.

#### 3.8.5. Time from Transplant

Twenty-seven studies investigated the time from transplant, and the weighted mean difference was 2.65 years, with a pooled standard error of 0.551. The pooled mean difference in time from transplant showed that (n = 1410) 42.6% of the 3309 responders had 2.65 more years from transplant compared to the non-responders (pooled *p* < 0.001), indicating that a longer time from transplant is associated with a higher vaccine response.

#### 3.8.6. Calcineurin Inhibitors

The weighted odds ratio of all 26 studies that represented calcineurin inhibitor use was 1.13. The calcineurin inhibitors used at the time of vaccine administration had no significant effect on the pooled odds of the vaccine response (pooled *p* = 0.651).

#### 3.8.7. Deceased Donor Status

Nine studies explored the status of a deceased donor as a risk factor. The weighted odds ratio was 1.005, showing that there was no significant effect on the pooled odds of the vaccine response (pooled *p* = 0.98).

#### 3.8.8. Gender

The weighted odds ratio of all 38 studies (n = 5067 total patients) considering gender was 1.016. Gender had no significant effect on the pooled odds of the vaccine response (pooled *p* = 0.8).

#### 3.8.9. mTOR Inhibitors

Among 25 studies and 3297 total patients, the weighted odds ratio of all studies investigating mTOR inhibitors was 1.87. mTOR inhibitors increased the pooled odds of the vaccine response by 87% (pooled *p* < 0.001), indicating that the increased use of mTOR inhibitors was associated with a higher vaccine response.

#### 3.8.10. Rituximab Use

The weighted odds ratio of all three studies taking Rituximab exposure into consideration was 0.325. Rituximab exposure had no significant effect on the pooled odds of the vaccine response (pooled *p* = 0.152).

#### 3.8.11. Lymphocyte Count

The weighted mean difference of all six k Lymphocyte count studies, with a total of 816 patients, was 0.288, with a pooled standard error of 0.081. The pooled mean difference in lymphocyte counts showed that the responders had increased lymphocyte counts (by 0.288 units) compared to the non-responders (pooled *p* < 0.001). A higher lymphocyte count is associated with a higher vaccine response.

## 4. Discussion

This systematic review and meta-analysis reported the available data on COVID-19 vaccines in the SOT population (Figure 6 and Figure 7). It highlighted the COVID-19 vaccine’s decreased immunogenicity after the initial doses of the Pfizer-BioNTech (BNT162b2), Moderna (mRNA-1273), Oxford/AstraZeneca (ChAdOx1 nCoV-19), and Janssen (Ad26.COV2.S) vaccines were administered [14,19,21,59,61]. The non-response rates varied widely, and they were higher for the Janssen (Ad26.COV2.S) COVID-19 vaccine [59]. Otherwise, the response rates were much higher than those after the first dose of an mRNA vaccine series [59]. On the other hand, the response rates among SOT recipients slightly increased after receiving second and third doses of a COVID-19 vaccine [33]. Most of the studies included in the analysis reported data on humoral responses and neutralization assays, although some of them suggested that the cellular responses were more effective than the humoral antibody responses were. Discrepancies between the studies’ results may have been due to the lack of standardization of the assays for detecting T cell immunity and the various performances across the different techniques. Data on immunogenicity after a fourth dose remain limited. Data regarding a fifth dose showed an additional benefit, leading to better serological responses. The studies suggested that patients who had detectable antibodies before receiving fourth and fifth doses of a vaccine showed significantly higher antibody concentrations compared to those who did not have any response to a vaccine [28]. The immunogenicity data suggested that there were higher response rates for the mRNA vaccines than there were for the adenovirus vector vaccines [41]. One study concluded that there were no significant differences between the inactivated whole-virus SARS-CoV-2 vaccine (Sinovac) second dose response rate versus the BNT162b2 BioNTech/Pfizer second dose response rate in transplant recipients [42].

The risk factors for low immunogenicity responses often included being of an older age and on immunosuppressive regimens consisting of MMF, glucocorticoids, calcineurin inhibitors, and belatacept [52,53]. In contrast, a higher seroconversion rate was linked to a higher eGFR, a younger age, and a longer period of time since transplantation [43,45,52,53]. In KT recipients, the mycophenolic acid (MPA) dose and hemoglobin level were found to be independent predictors of the antibody response, according to Tammy Hod et al. The probabilities of a favorable reaction were reduced by 17% when the MPA dose was increased by 1 mg/kg (odds ratio = 0.83; 95% CI = 0.75–0.92; *p* = 0.001). Additionally, it was discovered that the antibody response was reduced by 63% (*p* = 0.04) at hemoglobin levels of <13 g/dL [52]. Marta Kantauskaite et al. determined that KT patients had a diminished immunological response to the SARS-CoV-2 vaccine which was associated with the intensity of a recipient’s MMF treatment. Moreover, out of 225 KT recipients, 187 were administered MMF, and among them, 26 (13.9%) developed antibodies. Among the 26 seropositive KT recipients, 23 had an MMF dose of less than or equal to 1 g per day [53]. A failure to seroconvert following two-dose vaccination was found to be independently associated with patients who were 60 years of age or older (OR = 4.50; *p* = 0.02) and the use of anti-metabolites as immunosuppressive drugs (OR = 5.26; *p* = 0.004), according to Gianluca Russo et al. However, the seroconversion rate was high (92.9%) in younger patients who were not taking antimetabolites. [45]. Sebastian Rask Hamm et al. came to the conclusion that humoral non-response was related to the following factors: age (relative risk (RR) = 1.23 per 10-year increase, 95% CI = 1.11–1.35 and *p* = 0.001) within 1 year of transplantation (RR of 1.55, 95% CI 1.30–1.85, *p* = 0.001), treatment with MMF (RR = 1.54, 95% CI = 1.09–2.18 and *p* = 0.015), treatment with corticosteroids (RR = 1.45, 95% CI = 1.10–1.90 and *p* = 0.009), kidney transplantation (RR = 1.70, 95% CI = 1.25–2.30 and *p* = 0.001), lung transplantation (RR = 1.63, 95% CI = 1.16–2.29 and *p* = 0.005), and de novo non-skin cancer comorbidity (RR = 1.52, 95% CI = 1.26–1.82 and *p* = 0.001) [43].

This systematic review and meta-analysis has certain limitations. Most of the included studies were observational studies with limited sample sizes and few available randomized clinical trials. In addition to the lack of consistency of the assays assessing the cellular and humoral immunogenicity levels in the included studies, they were performed at different time points. Moreover, the heterogenicity in populations and the designs of the studies that were used to evaluate immunogenicity among SOT recipients limited our ability to perform a meta-analysis with the currently available data. Targeted strategies are needed to improve the cellular and humoral response rates in this population, such as additional COVID-19 vaccine doses, heterologous vaccinations, and lowered intensities for maintenance immunosuppression.

## 5. Conclusions

Solid organ transplant recipients are prioritized for immunization against COVID-19 due to the higher rate of developing COVID-19-related complications. Booster doses are highly encouraged to reach an adequate response rate among SOT populations. A variety of risk factors strongly limit the efficacy of the COVID-19 vaccines. To increase vaccine responses in this population, further approaches are urgently required in addition to carrying out carefully monitored clinical trials to assess the effect of vaccine boosters and modulation immunosuppressive regimens.

## Figures and Tables

**Figure 1 vaccines-11-01166-f001:**
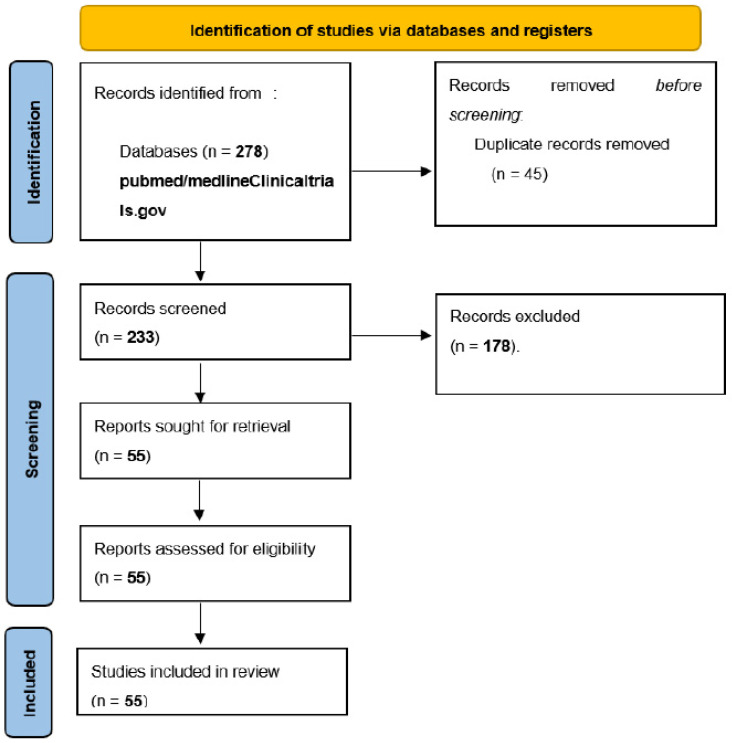
PRISMA flow chart of the included studies.

**Figure 2 vaccines-11-01166-f002:**
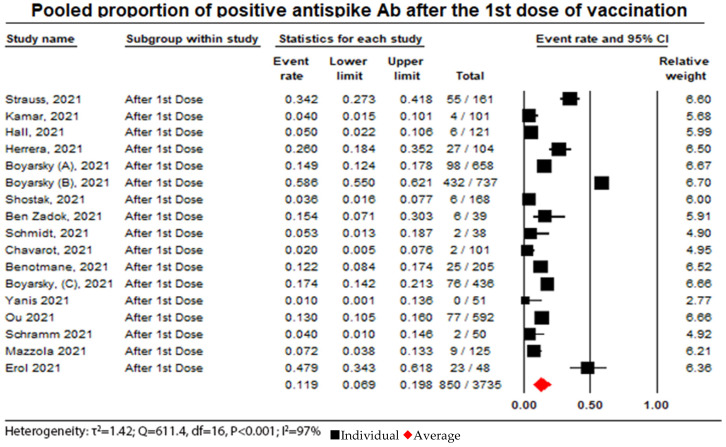
Pooled proportion of positive anti-spike Ab after administration of a first dose. Number of studies: 17. Number of patients: 3735. Number of responses: 850. Proportion (95% CI): 0.12 (0.07 to 0.2). The pooled proportion of positive anti-spike Ab after a first dose of vaccination from the random effects model was 0.12 (95% CI: 0.07 to 0.2). There was considerable heterogeneity between the studies reporting seroconversion rates after the administration of a second dose of the vaccine (I^2^ = 81.6%) [10,13,14,15,17,18,20,21,25,26,28,34,36,39,42,45,47].

**Figure 3 vaccines-11-01166-f003:**
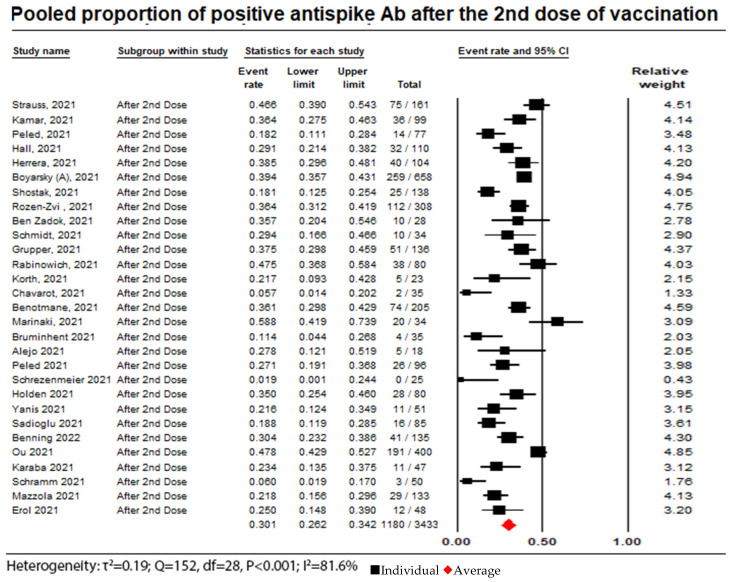
Pooled proportion of positive anti-spike Ab after administration of a second dose. Number of studies: 29. Number of patients: 3433. Number of responses: 1180. Proportion (95% CI): 0.30 (0.26 to 0.34). The pooled proportion of positive anti-spike Ab after a second dose of vaccination from the random effects model was 0.3 (95% CI: 0.26 to 0.34; N = 3433). There was substantial heterogeneity between the studies reporting on seroconversion rates after the administration of a second dose of vaccine (I^2^ = 81.6%). We noted that the seroconversion rate increased from 12% after the first dose to 30% after the second dose [10,12,13,14,15,18,19,20,21,22,23,24,26,27,29,30,32,33,34,35,36,37,38,39,40,42,44,45,47].

**Figure 4 vaccines-11-01166-f004:**
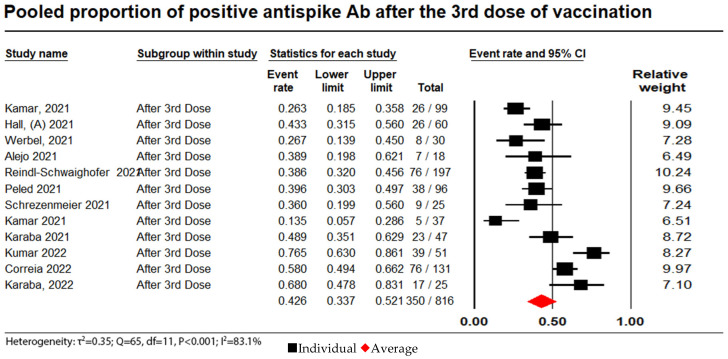
Pooled proportion of positive anti-spike Ab after administration of a third dose. Number of studies: 12. Number of patients: 816. Number of responses: 350. Proportion (95% CI): 0.43 (0.34 to 0.52). The pooled proportion of positive anti-spike Ab after a third dose of vaccination from the random effects model was 0.43 (95% CI = 0.34 to 0.52; N = 816). There was substantial heterogeneity between the studies reporting on seroconversion rates after the administration of a third dose of a vaccine (I^2^ = 83.3%) [6,11,16,30,31,32,33,34,40,41,46,60].

**Figure 5 vaccines-11-01166-f005:**
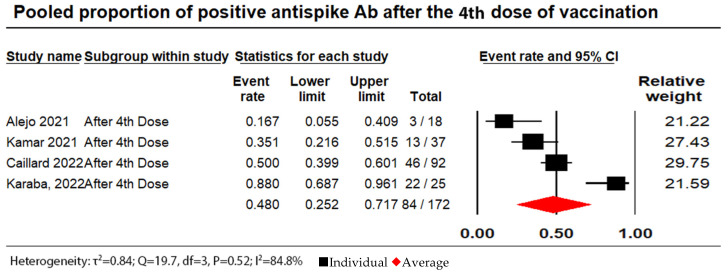
Pooled proportion of positive anti-spike Ab after administration of a fourth dose. Number of studies: 4. Number of patients: 172. Number of responses: 84. Proportion (95% CI): 0.48 (0.25 to 0.72). The pooled proportion of positive anti-spike Ab after a fourth dose of vaccination from the random effects model was 0.48 (95% CI: 0.25 to 0.72; N = 172). There was substantial heterogeneity between the studies reporting on seroconversion rates after the administration of a fourth dose of vaccine (I^2^ = 84.8%) [30,34,43,60].

**Figure 6 vaccines-11-01166-f006:**
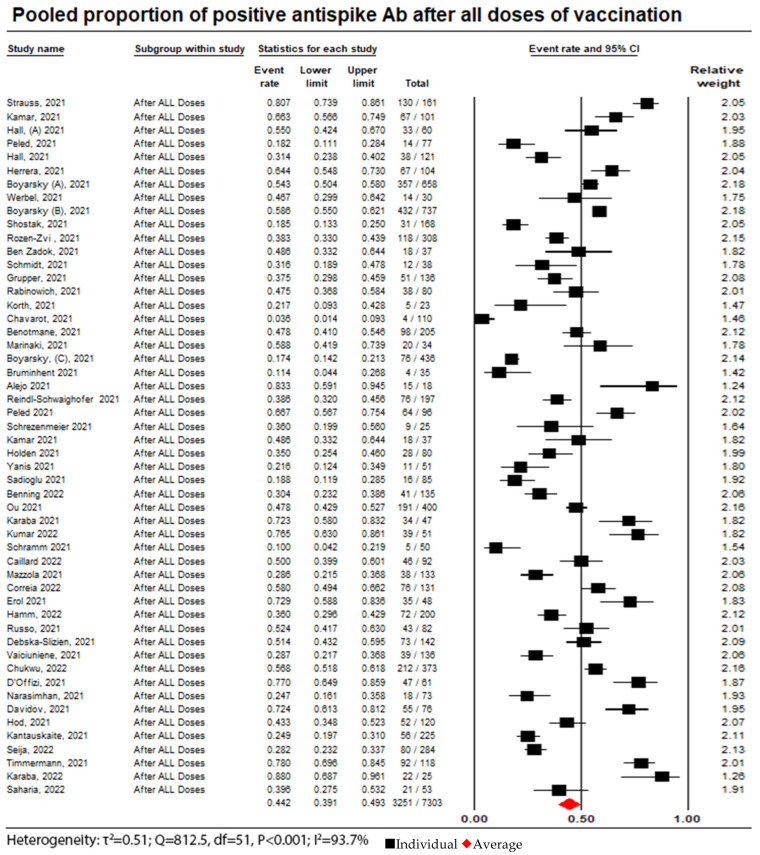
Pooled proportion of positive anti-spike Ab after administration of all doses. Number of studies: 52. Number of patients: 7303. Number of responses: 3251. Proportion (95% CI): 0.44 (0.39 to 0.49). The pooled proportion of positive anti-spike Ab after the administration of all doses of vaccination from the random effects model was 0.44 (95% CI: 0.39 to 0.49; N = 7303). There was considerable heterogeneity between the studies reporting on seroconversion rates after the administration of all doses of vaccine (I^2^ = 93.7%) [6,10,11,12,13,14,15,16,17,18,19,20,21,22,23,24,25,26,27,28,29,30,31,32,33,34,35,36,37,38,39,40,41,42,43,44,45,46,47,48,49,50,51,52,53,54,55,56,57,58,59,60].

**Figure 7 vaccines-11-01166-f007:**
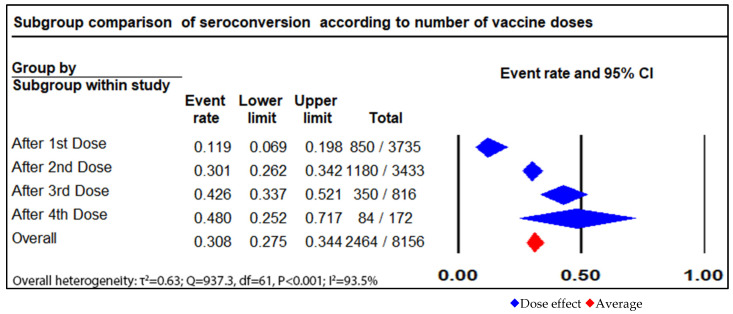
Different doses effect on seroconversion.

**Table 1 vaccines-11-01166-t001:** Reported factors that influence the immune response to vaccination.

Parameter	Effect Estimate	Lower Limit	Upper Limit	*p*	I^2^	Model	#Studies
Age *	−4.364	−5.673	−3.055	<0.001	44%	Random	29
Antimetabolites	0.238	0.183	0.309	<0.001	57%	Random	36
Antithymocyte globulin	0.847	0.573	1.251	0.404	0%	Fixed	6
BMI *	0.469	−0.898	−0.039	0.032	0%	Fixed	15
Time from transplant *	2.650	1.570	3.731	<0.001	78%	Random	27
Calcineurin inhibitors	1.130	0.665	1.922	0.651	79%	Random	26
Deceased donor status	1.005	0.651	1.552	0.980	54%	Random	9
Gender (males)	1.016	0.897	1.151	0.806	0%	Fixed	38
mTOR inhibitors	1.870	1.335	2.619	<0.001	44%	Random	25
Rituximab	0.325	0.070	1.511	0.152	0%	Fixed	3
Lymphocyte count *	0.288	0.130	0.446	<0.001	0%	Fixed	6

* Effect estimate, reported as the mean difference; all other parameters are reported as odds ratios.

## Data Availability

No new data were created or analyzed in this study. Data sharing is not applicable to this article.

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
