# Peer review of "Humoral and Cellular Immunity following Five Doses of COVID-19 Vaccines in Solid Organ Transplant Recipients: A Systematic Review and Meta-Analysis"

_vaccines, 2023, doi:10.3390/vaccines11071166_

Round 1

Reviewer 1 Report

It is an interesting topic for the study, the results are interesting but the discussion section needs to better explain the obtained results respecting the study limitations. More clear statements for the reader are needed. 

Please rewrite some sentences in abstract, because it is hard to read. 

…SOT recipients. And to identify factors… 

Despite the low immunogenicity of vaccines, additional strategies to improve vaccine response may be necessary.  – despite or due to?

Please reconsider how do we cite the authors, e.g. Marta Kantauskaite, et, al, determined …

needs major revision 

Author Response

We would like to express our sincere gratitude for your valuable time dedicated to evaluating our manuscript. We have taken into careful consideration all the concerns that were raised and made the necessary revisions accordingly.

  • The discussion and abstract have been revised.
  • All authors have been cited as recommended.
  • The conclusion part has been revised as supported by the results. 
  • The quality of the English language has been revised by using MDPI's English editing service (Certificate attached). 

Thank you for your time and consideration. 

Sincerely, 

Abdulmalik. 

Reviewer 2 Report

Peer review on: “Humoral and Cellular Immunity Following Five Doses of COVID-19 Vaccines in Solid Organ Transplant Recipients: A Systematic Review and Metanalysis”

by Abdulmalik Alotaibi et al.

General: Reference numbers in the text are not matching the reference list. This needs to be corrected.

Abstract:

1)    Spelling, grammar, and style need to be thoroughly revised. This refers mainly to the abstract and to the manuscript from chapter 3.6 ongoing.

2)    „We aim to assess the cellular and humoral immunogenicity (of different doses) of
COVID-19 vaccines in SOT recipients” – better: … (with increasing number of doses) …

3)    In case detailed information is given within the abstract regarding the publication selection process, major reasons for exclusions should be stated: “After title and Abstract screening, we retained 189 records, and 135 records were excluded.”

Methods:

4)    One important study (Haller et al, Transpl Int. 2022 Jan 4;35:10026. doi: 10.3389/ti.2021.10026) on serologic response rates after a 2-dose vaccination program (Moderna vs BioNTech) in 320 kidney transplant recipients is missing in this review and metaanalysis: What is the reason for excluding this work? Was it excluded being a “letter to the editor” without an available abstract? If so, the selection process seems inappropriate.

Results

5)    Figure A7: the line “overall” should be omitted in the forest plot, since it makes no sense.

6)    In-text citation style is uncommon and complicated: only (family name) et al, and not first/middle name initials and academic degree (e.g.: “Aura T. Abedon, BS, and
colleagues” should be written out

7)    Chapter 3.6: beyond describing the results of studies in various organ Tx recipients, it would be desirable to see within this meta-analysis a pooled effect of the type of organ (LiverTx vs KT vs HT, vs LungTx) on vaccination success. Does this meta-analysis confirm the increased risk for failure in KT and LungTx recipients, as compared to HTx, as revealed by Hamm SR et al ?

8)    Chapter 3.8 Risk factors influenced immunogenicity of COVID-19 vaccines: (See Table 1) – this chapter head needs to be revised

please, see above

Author Response

We would like to express our sincere gratitude for your valuable time dedicated to evaluating our manuscript. We have taken into careful consideration all the concerns that were raised and made the necessary revisions accordingly.

Regarding the abstract:

1- The quality of the English language has been revised by using MDPI's English editing service ( Certificate attached).

2- This point regarding the aim has been revised. 

3- The exclusion criteria are now added to the abstract.

Regarding the method point (4) and the study by Haller et al, it was excluded from our systematic review as it did not meet the predetermined inclusion criteria. Specifically, the sample population of that study had a history of previous exposure to COVID-19, which deviated from our inclusion criteria. Consequently, we made the decision to exclude the study from our analysis.

 Regarding the results: 

5- Figure A7 has been revised as suggested. 

6- The in-text citation style is revised.

7- Regarding the risk of failure among organs; as our primary objective did not include, nor the data collected involved comparing the risk of failure among different organs, we unfortunately would not be able to provide a specific analysis in our study results regarding this aspect.

8- The chapter head has been revised. 

Thank you for your valuable time and insights, 

Truly appreciated, 

Sincerely. 

Round 2

Reviewer 1 Report

I accept in the present form